# Cyclodextrin Complexation of Fenofibrate by Co-Grinding Method and Monitoring the Process Using Complementary Analytical Tools

**DOI:** 10.3390/pharmaceutics14071329

**Published:** 2022-06-23

**Authors:** Balázs Attila Kondoros, Ottó Berkesi, Zsolt Tóth, Zoltán Aigner, Rita Ambrus, Ildikó Csóka

**Affiliations:** 1Faculty of Pharmacy, Institute of Pharmaceutical Technology and Regulatory Affairs, University of Szeged, Eötvös Str. 6, H-6720 Szeged, Hungary; kondoros.balazs.attila@szte.hu (B.A.K.); aigner.zoltan@szte.hu (Z.A.); csoka.ildiko@szte.hu (I.C.); 2Faculty of Science and Informatics, Department of Physical Chemistry and Materials Science, University of Szeged, Béla Rerrich Square 1, H-6720 Szeged, Hungary; oberkesi@chem.u-szeged.hu; 3Department of Medical Physics and Informatics, University of Szeged, Korányi Fasor 9, H-6720 Szeged, Hungary; ztoth@physx.u-szeged.hu

**Keywords:** DIMEB, molecular complexation, solvent-free method, XRPD, DSC, FT-IR, SEM, dissolution studies

## Abstract

Solvent-free preparation types for cyclodextrin complexation, such as co-grinding, are technologies desired by the industry. However, in-depth analytical evaluation of the process and detailed characterization of intermediate states of the complexes are still lacking in areas. In our work, we aimed to apply the co-grinding technology and characterize the process. Fenofibrate was used as a model drug and dimethyl-β-cyclodextrin as a complexation excipient. The physical mixture of the two substances was ground for 60 min; meanwhile, samples were taken. A solvent product of the same composition was also prepared. The intermediate samples and the final products were characterized with instrumental analytical tools. The XRPD measurements showed a decrease in the crystallinity of the drug and the DSC results showed the appearance of a new crystal form. Correlation analysis of FTIR spectra suggests a three-step complexation process. In vitro dissolution studies were performed to compare the dissolution properties of the pure drug to the products. Using a solvent-free production method, we succeeded in producing a two-component system with superior solubility properties compared to both the active ingredient and the product prepared by the solvent method. The intermolecular description of complexation was achieved with a detailed analysis of FTIR spectra.

## 1. Introduction

Fenofibrate (FEN) is a widely used antidyslipidaemic drug that can lower blood cholesterol and triglycerides levels [1]. This active pharmaceutical ingredient (API) belongs to group II of the Biopharmaceutical Classification System (BCS) because of its low water solubility and high lipophilicity [2]. For substances in this group, the major reason for the extent of absorption in the gastrointestinal tract is their limited solubility and rate of dissolution from the formulation [3,4]. The low solubility of the drug can cause formulation problems and limited therapeutic efficacy, which can lead to bioavailability problems and increase the risk of dose-dependent side effects [5]. FEN has four known crystalline forms with different physical properties; three of them show low thermal stability [6].

Complexation with cyclodextrins (CDs) and CD derivatives is one of the outstanding solutions to improve the physicochemical properties of low solubility drugs. These cyclic oligosaccharides interact with drugs via dynamic complex formation and other mechanisms to mask undesirable physicochemical properties, including poor water solubility and limited stability [7,8,9]. Although natural CDs have been used broadly to improve pharmaceutical properties, the benefits of chemically modified CDs are superior in solubility, dissolution rate, and stability [10]. On the other hand, amorphous CD derivatives can be used to prepare amorphous complexes, which is also beneficial for achieving an increase in solubility [11]. Amorphous forms lack lattice energy and exhibit higher configurational energies than their crystalline forms, and provide better dissolution properties; therefore, amorphous properties are desired [12]. However, these non-crystalline materials present limited stability [13]. Ensuring the long-term stability of amorphous materials often encounters problems that are a major challenge in drug formulation work [14].

Conventional methods of forming a molecular complex with cyclodextrins generally use a defined amount of organic solvent [8]. The purpose of using the solvent mixture is to bring all or at least a small part of the components into the solution so that a secondary bond can be formed between the molecules which ensures adequate stability of the inclusion complex. Generally, the total amount of materials is dissolved but in the case of the so-called kneaded products, only a small amount of solvent mixture is used [7].

Recently the possibility of forming a molecular complex free of organic solvents has begun to be investigated in more detail. It was observed that in addition to the formation of the physical mixture, it was possible to produce inclusion complexes with more favorable properties with the help of the energy introduced during co-grinding [15]. Today, the industrial application of so-called “green technologies” free of organic solvents has come to the fore because they can reduce the cost of production and do not require the use of expensive analytical methods to determine the residual solvent content of the product [16,17]. Recently, several publications have been published in the literature investigating the possibilities of co-grinding cyclodextrin-containing molecular complexes [16,18].

Monitoring the production of cyclodextrin complexes is not an easy task and can only be performed by evaluating several complementary methods [19]. Thermal analysis by differential scanning calorimetry (DSC), supplemented by powder X-ray diffractometry (XRPD) studies, is essential in the investigation of putative complexes [20,21,22,23]. Furthermore, spectroscopic methods such as Fourier-transform infrared spectroscopy (FT-IR) can offer an important supporting tool. Finally, scanning electron microscopy (SEM) measurements have been shown to be useful for explaining and better understanding solid phase changes [24,25,26,27].

Aigner et al. have previously studied the formation of the FEN cyclodextrin complexes. Physical mixtures, kneaded products, and precipitated products were prepared at different molecular ratios with different CD derivatives, and the dissolution rate and in vitro membrane diffusion properties of the products were investigated. It was found that these in vitro properties depended on the method of product preparation and the molecular ratio. Appropriate improvements in physicochemical parameters could also be achieved for physical mixtures. Best results and solubility increase were achieved for heptakis-(2,6-di-O-methyl)-β-CD (DIMEB) [28]. Studying the complex formation requires different analytical procedures, and their mutual evaluation is necessary to explore the process [29].

Based on preliminary studies detailed above in the present paper, we aimed to produce the FEN:DIMEB molecular complex using organic solvent-free technology and characterize the changes in physicochemical properties over time in the process. Various analytical methods were used to monitor physicochemical changes during co-grinding, such as DSC, and XRPD studies to evaluate crystallinity and thermal properties, and SEM for morphological changes. The time course of intermolecular interactions was studied via FT-IR with correlation analysis. Co-ground products were compared to the kneaded products in in vitro dissolution and diffusion studies.

## 2. Materials and Methods

### 2.1. Materials

FEN (isopropyl-2-[4-(4-chlorbenzoyl)phenoxy]-2-methylpropanoat) was kindly donated by Chemical Works of Gedeon Richter Plc. (Budapest, Hungary). Heptakis-(2,6-di-O-methyl)-β-CD (degree of substitution: 14; isomeric purity: 95%; molar mass: 1331.0 g mol^−1^) was obtained from Cyclolab R&D Laboratory Ltd. (Budapest, Hungary). Simulated intestinal fluid (pH 6.8), without enzymes, was prepared based on the monograph 5.17.1 of the European Pharmacopeia (10th edition). A 77.0 mL amount of 0.2 M NaOH, 250.0 mL of a solution containing 6.8 g of KH_2_PO_4_, and 500 mL of water were mixed; pH was adjusted to pH 6.8 and diluted to 1000 mL with water. All used materials for this solution were purchased from Sigma-Aldrich (Budapest, Hungary).

### 2.2. Methods

#### 2.2.1. Preparation of Co-Ground and Kneaded Products

The FEN:DIMEB two-component products with a 1:1 molecular ratio were prepared by co-grinding and kneading. As the goal was to produce a low-weight, homogeneous product, during co-grinding an agate mortar was used. The duration of co-grinding was determined to be 60 min based on the results of the preliminary tests. An appropriate amount of sample was taken every 5 min for physicochemical investigations.

In the kneading preparation method of the inclusion complex, a minimum amount of 50% (*v*/*v*) of ethanol-water mixture was used. The product was also made in a mortar. After the addition of the liquid phase, the suspension system was stirred continuously until most of the liquid had evaporated. The product thus obtained was dried in a vacuum drier for 24 h at room temperature and then carefully pulverized.

The products were stored under normal conditions in a screw-capped glass vial for the duration of the assays.

#### 2.2.2. Differential Scanning Calorimetry

DSC analysis of each component and product was performed using a Mettler Toledo DSC 821^e^ (Mettler-Toledo GmbH, Greifensee, Switzerland) using STARe software (version 9.30, Mettler-Toledo GmbH, Greifensee, Switzerland). The instrument was calibrated using indium. During the tests, the heating rate was 5 °C min^−1^, and 10 L h^−1^ of Argon gas was used as the purge gas. In our studies for different purposes, we used several temperature ranges (25–140 °C; 25–110 °C, followed by recooling and reheating to 140 °C). The weight of our test samples ranged from 2 to 5 mg in each case. Assays were performed using a 40 μL covered aluminium sample holder with 3 holes.

#### 2.2.3. X-ray Powder Diffractometry

XRPD measurements were performed with a BRUKER D8 Advance diffractometer (Bruker AXS GmbH, Karlsruhe, Germany) equipped with a Våntec-1 line detector system with Cu KαI radiation (λ = 1.5406 Å) over the interval 3–40°/2θ. The measurement parameters were as follows: target, Cu; filter, Ni; voltage, 40 kV; current, 40 mA; time constant, 0.1 s; angular step 0.010°. A low-background silicon sample holder was used in our studies.

#### 2.2.4. Infrared Spectroscopy

The attenuated total reflectance (ATR) spectra of both starting materials and samples were taken by increasing grinding time. They were measured, on a Bio-Rad Digilab Division FTS-65A/896 FT-IR spectrometer (Bio-Rad Digilab Division, Philadelphia, PA, USA) equipped with a liquid nitrogen-cooled Mercury-Cadmium-Telluride (MCT) detector, at 4 cm^−1^ optical resolution, between 4000 and 400 cm^−1^. A Harrick’s Meridian SplitPea single reflection diamond ATR accessory was used. To achieve a proper signal-to-noise ratio, 1024 interferograms were averaged. Since the efficiency of the dry air purge of the internal part slightly improved during the measurement of spectra, the single-beam spectrum of the empty beam prior to and at the end of the measurements was also saved, and the absorbance spectrum of the mixture of water vapor and gaseous carbon dioxide was calculated.

Spectra were exported to Thermo Scientific’s SPC format by Digilab’s WinIR vers. 3.2 program (Bio-Rad Laboratories, Inc., Cambridge, MA, USA), controlling the spectrometer. All further data manipulations were performed by Thermo Galactic’s GRAMS/AI vers. 7 program (Thermo Fisher Sciencific Inc., Waltham, MA, USA).

#### 2.2.5. Correlation Analysis

Spectroscopic data were analyzed by the generalized two-dimensional (2D) correlation method developed by Noda et al. [30,31]. A program used to perform the analysis was written by Szabó et al. [32] in the internal program language of GRAMS/AI (Array Basic).

ATR spectroscopy cannot provide spectra with comparable intensities since those depend on several factors including the number and the size of the particles pressed to the active element of the ATR accessory. So, all spectra were treated prior to the 2D correlation analysis, in order to avoid false results [33]. First, the peaks of water vapor, in the 1700–1500 cm^−1^ region, were removed by subtracting the spectrum calculated from the single beam spectra of the empty beam, mentioned earlier. Secondly, the spectra were truncated to the 1800–630 cm^−1^ region, which contained the peaks with reliable intensities. Thirdly, slight linear baseline correction took place using the following points 1800, 1530, 1210, 875 and 630 cm^−^^1^, to remove the fluctuations in the baseline curvature. After an offset correction between 1800 and 1530 cm^−^^1^, the intensities of the spectra were normalized to the most intensive peak of DIMEB at 1047 cm^−^^1^, where there were no peaks of FEN. Finally, five points, between 673 and 665 cm^−^^1^, were zapped from the spectra, to remove the peak at 667 cm^−^^1^, belonging to the Q-branch of the bending mode of gaseous carbon dioxide.

The correlation analysis program supplied three files as output, each in a multifile. SPC format. The dynamic surface was calculated first, using the average spectrum as a reference, in each run. The synchronous and the asynchronous correlation surfaces were calculated from it in the second step of each run. GRAMS3D of Thermo Galactic (Thermo Fisher Scientific Inc., Waltham, MA, USA) was used for the visualization of the three-dimensional (3D) surfaces.

#### 2.2.6. Scanning Electron Microscopy

A Hitachi S-4700 field emission cathode scanning electron microscope (FESEM) was used to capture electron images. The X-ray emission from the samples was detected with an integrated Röntec QX2 EDS (Energy Dispersive X-rays spectroscopy) device. The accelerating voltage of the electrons was 20 kV. The grains from the samples were fixed on conducting carbon tape. The samples were not covered by any conducting layer, in order to maintain an undisturbed X-ray signal from the grains. Secondary electron images were recorded in low magnification, fast scan mode, to avoid the local charge accumulation in the samples. For detection of the elemental composition, X-ray spectra from the grains were recorded. Element mapping was performed as well, to visualize the spatial distribution of the constituents.

#### 2.2.7. In Vitro Dissolution Rate Studies

Dissolution studies were performed using the rotating paddle method in the Pharmatest PTW II apparatus (Pharma Test Apparatebau AG, Hainburg, Germany). An amount of 8.88 mg FEN and products containing the same amount of FEN were placed in 50 mL of simulated intestinal fluid (pH 6.8). The dissolution medium was thermostated at 37 °C and stirred at 100 rpm. Samples were taken at predetermined intervals (5, 10, 20, 30, 60, 90, 120 min). All aliquots were immediately filtered (0.45 μm pore size syringe membrane filter) and at each sampling time, the dissolution medium was replaced with an equal volume of simulated intestinal fluid. Samples were measured spectrophotometrically at 296 nm using Unicam UV-Vis spectrometer (Unicam Ltd., Waltham, UK). Mathematical correction was applied for the cumulative dissolution caused by replacing samples with fresh medium.

The cumulative dissolution was performed by taking into account the exchange of dissolution medium during sampling. Dissolution efficiency (DE) calculations were performed based on the values measured on the dissolution curves at 60 and 120 min, using the following equation:(1)$$\mathrm{DE}=\frac{\int_{0}^{t}\mathrm{ydt}}{y_{100}}100\%$$
where y and y_100_ are the cumulative percentage dissolution at time t and 100% dissolution, respectively.

#### 2.2.8. In Vitro Diffusion Studies

Diffusion studies were carried out for broader in vitro characterization. A 8.88 mg amount of FEN and the products with the same amount of API were dissolved in 50 mL of simulated intestinal medium; 4 mL of this solution was placed in dialysis bags (Spectra/Por^®^, Spectrum Laboratories Inc., Rancho Dominguez, CA, USA). The bags were placed immediately in phosphate buffer (pH 7.4) as the acceptor phase (representing blood) in all cases. The system was thermostated at 37 °C and stirred at 100 rpm. In predetermined time intervals, 2 mL of samples were taken and exchanged with fresh medium. API concentration was measured at 296 nm using Unicam UV-Vis spectrometer (Unicam Ltd., Waltham, UK).

## 3. Results and Discussion

### 3.1. Differential Scanning Calorimetry

Differential scanning calorimetry was performed with the starting materials and the products obtained by co-grinding.

A sharp endothermic peak was observed on the FEN thermogram at ca. 80.5 °C, which corresponded to the melting point of API. The test does not suggest the decomposition of the molten material at elevated temperatures. On the thermogram of the DIMEB, a wide, flat endothermic peak is observed between 25 and 70 °C, caused by the moisture content of the material. Above 70 °C no additional thermoanalytical signal can be detected. Since the endothermic peaks of DIMEB and FEN do not overlap, the thermoanalytical changes caused by co-grinding can be well observed (Appendix A).

The thermogram of the physical mixture (0 min of grinding) of FEN and DIMEB shows the endothermic peak caused by the melting point of the drug. The peak broadened due to the presence of the complexing component, and the area under the curve decreased according to the weight ratio of the two-component product (Figure 1). As the grinding time increased, the area under the curve corresponding to the melting point of the drug decreased, which means a continuous decrease in the amount of crystalline drug. With a co-grinding time of 60 min, the presence of the crystalline active ingredient is already difficult to detect. The DSC curve does not provide information on whether amorphization or molecular complexation during co-grinding caused a decrease in the area under the curve of the endothermic peak. To confirm this, the results of further studies (XRPD, FT-IR) must also be considered.

When plotting the normalized integral values as a function of co-grinding time, it can be seen that the decrease in the endothermic peak can be described by two different stages. In the first 20–25 min, a rapid linear change occurs, and then after the appearance of the exotherm peak, this change slows down. The growth rate of the exothermic peak can be described by a process (Figure 1A).

In addition to the endothermic peak with a decreasing area that could be assigned to the melting point of FEN, the appearance of a new exothermic peak was observed from the 20th minute of the co-grinding time at a temperature higher than the melting point temperature. The area under the curve of the exothermic peak increased with increasing grinding time and the peak shifted slightly towards the higher temperature range (from ca. 90 to 96 °C). Based on the thermogravimetric investigations performed as a control, there was no change in mass in this temperature range for the two-component products, so it is assumed that the exotherm signal is caused by an internal phase transition (Figure 1B).

A complex heating procedure was used to interpret the new exotherm peak that appeared. The product co-ground for 60 min was first heated to 110 °C, cooled, and reheated to 140 °C. No thermoanalytical signal was observed during cooling and the second heating, confirming that the product formed during the first heating has adequate stability, in which no internal phase transformation takes place during the second heating (Figure 1C).

### 3.2. XRPD

Diffractograms of the starting materials and the co-ground products were recorded. FEN was a crystalline material with sharp characteristic peaks, polymorph I according to the Cambridge Structural Database, and DIMEB was a material with a completely amorphous structure.

The physical mixture of FEN and DIMEB contains drug-specific peaks that are superimposed on the amorphous diffractogram of CD. The intensity of the peaks corresponds to the FEN content of the two-component product. During 60 min of co-grinding, the intensity of the peaks decreases continuously, and the product becomes amorphous. This observation supports the continuous decrease and disappearance of the endothermic peak at the FEN melting point observed in DSC studies. However, it does not provide an answer as to whether only amorphization occurred or whether the formation of a molecular inclusion complex caused the characteristic peaks to decrease and then disappear (Figure 2)

It is noted that by repeating the XRPD tests of the co-ground products stored under normal conditions after 2 years of storage, completely identical curves were obtained, which show the outstanding stability of the samples (Appendix A).

### 3.3. Evaluation of IR Spectroscopic Data

The alterations caused by the grinding process resulted in very complex changes in the ATR spectra of the samples taken. So, no simple methods, e.g., succeeding spectral subtraction, or even more sophisticated methods like Fourier deconvolution or peak fitting, were able to provide a reliable basis for the understanding of the processes that occurred during grinding.

The dynamic surface constructed from the normalized spectra, shown in Figure 3, gave the clue that the grinding process should be studied in three 20-min long periods.

The intensity alterations indicated that the behavior of the peaks could be classified into three groups by the grinding time. There were peaks with increasing intensities, others just did the opposite, and some had a maximum around the middle of the grinding time. The last behavior suggested that processes were having just the opposite influence around the maxima.

The first and the last 20 min seemed to be dominated by a single main process, while the middle 20 min suggested a mix of processes, so they were studied accordingly.

The results of the correlation analysis were given in four spectral ranges defined earlier in the section describing the data treatment of the spectra.

#### 3.3.1. The First 20 min of Grinding

The first part of the synchronous correlation surface, which has plane symmetry, between 1800 and 1530 cm^−1^, is shown in Figure 4. This spectral region contained the peaks belonging to the C=O stretching vibrations of FEN and some IR active modes of the aromatic rings [34,35]. The diagonal peaks of the synchronous surface indicated that three peaks were changing in a concerted fashion. The positive sign of the six off-diagonal peaks showed that the alteration took place in the same direction. Their intensities indicated a strong correlation. The direction of the intensity alteration was given by the examination of the dynamic surface of the same region, shown in Figure 3. The axial sections of the surface, taken at 1728 cm^−1^ (A) and 1651 cm^−1^ (B) in Figure 4, showed that practically only the peak intensities of FEN were affected by the process. The intensities of these peaks increased during the first 20 min of the grinding process relative to the most intense peak of the DIMEB at 1047 cm^−1^.

Extending the examination to the whole synchronous correlation surface showed that it is dominated by peaks of positive correlation with the peaks discussed above, except for some peaks in the range nearby the peak of DIMEB, at 1047 cm^−1^, to which the normalization was made. That part of the dynamic surface (shown in Figure 3) confirmed that the relative intensity changes were just the opposite, decreasing. On the other hand, those diagonal peaks had off-diagonal peaks with a positive sign, indicating that their intensity changed in the same direction.

It was unexpected that during the first 20 min of grinding, the main changes in the recorded spectra were the increase of the relative peak intensities of FEN and the decrease in those of DIMEB. An obvious explanation could be given if the origin of the spectral information were taken into account in the case of ATR spectroscopy. The intensities measured by ATR spectroscopy are dependent on several factors [36]. One of the most important is the size of the area of the sample, in contact with the ATR crystal. This area is influenced by the size of the particles. The smaller the particles, the bigger the contact area. The studied FEN was in a crystalline state with macroscopic particles, while DIMEB was an amorphous material with much smaller aggregates prior to the grinding process, so it occupied a higher portion of the surface of the diamond crystal. The decreasing size of FEN caused increased contact area, a higher proportion on the surface, causing increased intensities (See Appendix A for graphical interpretation).

The asynchronous correlation surface, having axial symmetry, consists of only off-diagonal peaks. Analysis of it is usually not as straightforward as that of the synchronous one. The main reason is that intensity changes that are not parallel or antiparallel in the dynamic surface are collected into this surface. Features, like noise [37], or features occurring due to random events during sample preparation or measurement, are superimposed on the features caused by real asynchronous changes in the samples.

The asynchronous correlation surface gained from the spectra of the first 20 min of grinding is shown in Figure 5. The strongest correlation peaks occurred in the ranges around the characteristic peaks of FEN, but unexpectedly strong off-diagonal peaks were found in the regions where there were no strong bands either in the spectrum of FEN or in that of DIMEB. Some of them were marked with broken black arrows, on both sides of the diagonal line, in the below figure.

More than 20 axial sections of the surface, at the wavenumbers of the unexpected correlations, were studied. They all proved to be the same in shape, except in their sign, which depended on the sign of the off-diagonal peak they belonged to. There was another common feature in them: they were defined by a peak of no more than 3–5 points wide on the surface, suggesting that they might originate from a gaseous substance. Their position confirmed that they originated from the slight difference in the peak positions belonging to the water vapor between the spectra of samples and the spectrum of the empty beam used for the subtraction during the data preparation. These residual intensities were expected to show noise-like behavior.

One of the most intensive peaks was found at the X = 1726 cm^−1^, Y = 1732 cm^−1^ position in this part of the surface. Since it was an off-diagonal peak, both *X*- and *Y*-axis sections had to be studied, to find the spectral features affected by the asynchronous change in the samples. The corresponding sections were compared to the spectrum of FEN in Figure 5. The *X*-axis section indicated a strong correlation at 1726 cm^−1^ somewhat below the C=O stretching vibration of the ester group of FEN. Identification of other peaks was impossible using this section due to the interference of the water vapor peaks. The *Y*-axis section showed a strong correlation at 1632 cm^−1^, somewhat above the C=O stretching vibration of the ester group of FEN. Furthermore, it revealed that the other characteristic bands of FEN were also affected by the asynchronous change in the samples. Since the latter peaks were over the diagonal, the sign of the peaks should be reversed, meaning that the alteration influenced the C=O stretching band around 1653 cm^−1^ and the skeletal stretching bands of the rings around 1600 cm^−1^, followed by the alteration of the C=O stretching vibration of the ester group.

The corresponding synchronous surface showed a strong correlation at 1728 cm^−1^, from which these wavenumbers differed within the range of optical resolution, 4 cm^−1^, of the spectra, so the bands causing it strongly overlapped and prevented the successful application of other, more conventional methods. At the same time, these changes were superimposed on the synchronous change of the peak intensity at 1728 cm^−1^, so they have random-like behavior due to the heterogeneous state of the samples. Against this, the position of these peaks suggests that the environment of the ester group of some FEN molecules changed in a way that the C=O stretching band shifted to lower energies, indicating stronger interaction with its environment than that which was within the crystal frame, while this environment changed later, causing a shift of the same band to higher energy values, causing the asynchronous widening of the band.

Extending the study of the asynchronous correlation surface, the most significant features showed up in the range from 1320 cm^−1^ to 1047 cm^−1^, shown in Figure 6.

Most of them were attributed to the characteristic bands of FEN, marked with red arrows. Their behavior is opposite to the low energy side of the C=O stretching mode of the ester group. The number of these bands was assigned to the deformation modes of the methyl groups [38] and the C-C-O stretching modes of the ester group and the etheric oxygen between the aliphatic part and the non-chlorine substituted ring of the molecule [34], indicating a rearrangement in the aliphatic part of the molecule.

On the other hand, some peaks were assigned to the aromatic rings and even to the C-Cl stretching modes combined with the ring deformation modes at 1088 cm^−1^ or the in-plane bending mode of the O-substituted ring at 1183 cm^−1^ [34], indicating that the asynchronous processes also influenced the state of the rings.

There were certain broad features, like the most prominent one, centered around 1228 cm^−1^, on the surface which cannot be attributed either to FEN or to DIMEB, marked with a red ellipsis and purple dashed arrows. The formation of a new species from the interaction of FEN and DIMEB, besides the dispersion of FEN, might be the explanation.

#### 3.3.2. The Second 20 min of Grinding

The synchronous correlation surface generated from the IR spectra, recorded between 20 and 40 min of grinding, showed a completely different view than that of the first 20 min, in the range of 1800–1530 cm^−1^, (Figure 7).

The number of diagonal peaks increased to five from three. The axial sections taken at the corresponding peak positions, e.g., at 1740 cm^−1^ (A) and 1726 cm^−1^ (B) in Figure 7, also showed that the C=O stretching peaks were shifted from the peak positions of FEN, either to higher or to lower wavenumbers.

The other main difference from the previous period of grinding time was that there were also negative off-diagonal peaks in this range, also indicated by the axial sections of the surface. Examination of the dynamic surface confirmed that the positive off-diagonal peak indicated increasing intensity, while the negative peak meant decreasing intensity during the second period of grinding. Since the intensity of the diagonal peaks, which had a positive correlation, was higher in intensity, it was obvious that they indicate the main process of the period.

The following procedure was used to gain spectroscopic backing of the nature of this process. A normalized average spectrum was calculated from all spectra recorded using the samples taken during grinding. This spectrum contained all peaks of all species that were present in the samples, including the starting materials. So, successive subtraction of the spectra of FEN and that of DIMEB was performed to get the spectrum characteristic on the “PRODUCT” of grinding. The spectrum gained this way was compared to the axial cross-sections and the spectra of FEN and DIMEB, as shown in Figure 7. It was clear that the main process in the second 20 min of grinding was the formation of the “PRODUCT”, a complex between FEN and DIMEB, which showed the characteristic peaks of FEN and DIMEB at shifted positions due to the interaction between them. On the other hand, the negative synchronous correlation originated from the consumption of FEN during the above process.

A similar comparison, along with the full range of surfaces studied, verified the previous conclusion.

The asynchronous correlation surface gained from the spectra, recorded during the second 20 min long period of time, was clearer than that of the first 20 min. No strong contribution from water vapor or spectral noise was seen in the range of 1800–1530 cm^−1^, as shown in Figure 8. The wavenumbers gained from the position of the off-diagonal peaks were practically the same as those of the diagonal peaks on the synchronous surface. The x- and y-axial sections taken in the C=O stretching region of the ester group, (A) at 1740 cm^−1^ and (B) at 1728 cm^−1^, confirmed that the opposite sign depended on the “origin” of the peak, and they were opposite if they came from FEN or “PRODUCT” (Figure 8).

Examination of the full surface showed similar features, except where the corresponding peaks of FEN and DIMEB were too close to each other and resulted in an S-shaped feature in the axial sections of the surface.

This behavior was explained by the study of the dynamic surface, which showed a maximum in the ranges of the C=O stretching bands of FEN. At the beginning of the second 20 min long period of time, the influence of the dispersion of FEN was caused by grinding, but the formation of “PRODUCT” took over and the intensities at the peaks of it started growing faster, while the peak intensities of FEN started diminishing. So, the formation of “PRODUCT” required a certain small size of the FEN crystallites.

#### 3.3.3. The Last 20 min of Grinding

The synchronous correlation surface generated from the IR spectra, recorded between 40 and 60 min of grinding, were different either from that of the first, or that of the second 20 min, in the range of 1800–1530 cm^−1^ (Figure 9).

The diagonal peaks belonging to the skeletal modes of the aromatic rings around 1600 cm^−1^ nearly completely disappeared from the surface. The four strong diagonal peaks in the range of the C=O stretching modes of FEN switched intensity. The lower energy component became more intensive and had positive off-diagonal peaks, while the higher energy components were less intensive, and had negative off-diagonal peaks. Examination of the corresponding dynamic surface showed that this time the positive off-diagonal peaks meant decreasing intensity and the negative ones increasing intensity. A comparison of the axial sessions of the surface taken at the maxima of the most intensive peaks, shown in Figure 9, showed that the dominating process of this period of grinding was the consumption of the residue of FEN resulting in the formation of a small amount of “PRODUCT”. Examination of the other parts of the synchronous correlation surface confirmed this.

The asynchronous correlation surface of the last 20 min of grinding, shown in Figure 10, was very similar to that of the second 20 min (Figure 8) except that the peaks switched their sign like in the case of the asynchronous surface, and the bands of the aromatic ring were less influenced. It was also noted that the intensities of the peaks belonging to the aromatic ring stretching modes, around 1600 cm^−1^, had comparable intensities to those of the C=O stretching modes. The applicable dynamic surface showed the leveling-off of the synchronously increasing band intensities of “PRODUCT” at the end of the grinding period, as was expected on their sign.

Unfortunately, the other wavenumber ranges of the asynchronous correlation surface showed noise-like features, which were impossible to interpret.

#### 3.3.4. Molecular Considerations Based on the Spectra of FEN, DIMEB, and “PRODUCT”

Macrocrystalline Form-I of fenofibrate was used for the co-grinding experiment. Its infrared spectrum is between 1800 cm^−1^ and 630 cm^−1^, shown in Figure 11—purple spectra [6,39]. The available assignments of peaks were collected from the literature, and a summary of them is given in Appendix A [34,35,39,40,41]. Similar articles on cyclodextrins (CDs) were scarcely found [42,43,44,45].

Spectra of FEN, DIMEB, and “PRODUCT” are also compared in Figure 11 to understand the molecular aspects of the changes revealed by the correlation analysis.

The 1800–1530 cm^−1^ spectral range is dominated by the peaks of FEN in the time-averaged spectrum of “PRODUCT” (Figure 11A). Peaks were shifted and widened due to the changes in the interactions of the molecule. The C=O stretching mode of the ester group showed the most complex picture. Lowering the wavenumber of the band, which was present in all stages of grinding in the asynchronous surface, indicated that temporarily the strength of the interaction of the group increased in the process, compared to the crystalline state. On the other hand, the higher wavenumber component of the same mode showed that later the group got into an environment with a weaker strength of interaction than in the crystalline state, where it had three short moments of contact with the neighboring molecule [46]. Only the latter effect was seen in the case of the keto group between the aromatic rings, which had two short moments of contact in the crystal lattice [46]. The band system characteristic on the aromatic rings, around 1600 cm^−1^ and somewhat below—vibrations *8a* and *8b* according to Wilson’s notation [40,41]—widened and simplified.

The obvious explanation, at the molecular level, could be given, if the steric availability of the C=O groups was taken into consideration. The ester group is in the middle of a flexible aliphatic chain, while the keto group between the aromatic rings is in a hindered position since even the planar arrangement of the rings and the keto group cannot be achieved according to the theoretical calculations due to the repulsion between the hydrogens of the rings [34,35]. So, the ester group can serve as an anchor during the early stages of the formation of the FEN-DIMEB complex to provide the >C=O∙∙∙H-O- bond with the surface O-H groups of DIMEB, which helps it to get away from the crystal lattice. The hydrogen bridges meant stronger interaction, so the position of the C=O stretching peak must shift downward. Later, when the molecule got away from the crystal lattice and a FEN-DIMEB complex was formed, this bridge broke up, and other interactions took over and they kept the complex together. The strength of the interactions lessened for both C=O groups.

The range between 1530 and 1210 cm^−1^ showed a much more complex picture (Figure 11B). The spectrum of the “PRODUCT” showed bands both from FEN and DIMEB, but in shifted positions. The aromatic ring vibrations *19a* and vibration *14*—characteristic only of the chlorine-substituted aromatic ring—shifted upwards and retained their relative intensities. Vibrations 3 of the aromatic rings also shifted and retained their intensity, but the contribution from one of DIMEB’s C-O-H + CH3/CH wagging modes required a peak fitting procedure to reveal the direction. Ring vibrations *3* unexpectedly shifted downwards. Vibrations *19b* were impossible to identify. Most of the bands between 1400 and 1360 cm^−1^, characteristic of the iso-propyl and 2,2-dimethyl-propyl groups of FEN became hardly recognizable. On the other hand, the band at 1249 cm^−1^—characteristic of the O-C-C stretching mode of the ester group [34]—retained its intensity and shifted to 1253 cm^−1^. The characteristic bands of DIMEB are also present in the spectrum of “PRODUCT”. The most obvious broad band around 1460 cm^−1^ belongs to the deformation modes of the methyl and methine groups. The “fine structure” of this broad feature in the spectrum of “PRODUCT” was caused by the superposition of the corresponding modes of FEN present in the FEN-DIMEB complex. The other set of bands—assigned to the various combinations of O-H bending and the methyl and methine wagging modes by Sabapathy et al. [43]—were shifted to higher wavenumbers.

The range between 1210 and 630 cm^−1^ was dominated by somewhat modified bands of DIMEB with a minor contribution from FEN (Figure 11C,D). However, most of the peaks—characteristic of FEN, including vibrations *9a*, *18b*, and *18a* of the aromatic rings, and the peak at 1088 cm^−1^ assigned to the C-Cl stretching mode [34] and the methyl bending modes of iso-propyl and 2,2-dimethyl-propyl groups—disappeared, and the spectrum of “PRODUCT” shows the characteristics of DIMEB, but with shifts and the relative intensity changes of the bands. This was clear evidence that a new species, the FEN-DIMEB complex, formed. One of the few exceptions occurred at 763 cm^−1^, where the *6a* mode of the chlorine-substituted aromatic ring was expected in the spectrum of FEN, and an unassigned, probably wagging mode of the pyranose ring, was found in the spectrum of DIMEB. The band shifted to lower wavenumbers like all the bands of DIMEB in this range but inherited the shape of the peak of FEN. The shape of the peak resisted repeated subtraction attempts, but it could be a kind of artifact, like all weak bands below it. The peaks of the out-of-plane vibrations of the aromatic ring *17b* disappeared; the others lost their intensity, e.g., *4*.

Observations, analyzed above, suggested the conclusion that although hydrogen bonding between the C=O group of FEN and the -O-H groups of DIMEB plays a role in the formation of the FEN-DIMEB complex, this interaction is not present in the final product; rather, an inclusion complex formed, and at least one of the aromatic rings of FEN slipped into the hydrophobic internal space of DIMEB [47].

### 3.4. Scanning Electron Microscopy

The samples taken at 0, 20, and 60 min of grinding were also studied with SEM at various magnifications from 50 to 2000, and their EDS spectra were also recorded. The SEM images of the pure materials at magnification of 50, shown in Appendix A, clearly showed their difference in crystallinity. FEN contained crystals of various sizes from 200 μm downwards with well-defined sharp edges. The SEM image of DIMEB, taken at the same magnification, showed mainly aggregates of much smaller primary particles which were possible to see only at a magnification of 500.

SEM images of samples taken at 0 and 20 min of grinding time at a magnification of 200 were shown in Figure 12A,B. The image of the mixture of the starting materials showed that the DIMEB particles were either attached to the surface of the FEN crystals or were separately distributed between them. On the other hand, after 20 min of grinding, the ratios of the separated, small-sized DIMEB aggregates were much less, and FEN was much more evenly distributed and occupied a higher ratio on the surface of the bigger, mixed aggregates.

### 3.5. In Vitro Dissolution Studies

Dissolution studies were carried out using modified pharmacopeia methods in simulated intestinal fluid. The dissolution curves of the pure drug and its products with an equimolar ratio are shown in Figure 13 and corresponding calculated DE values in Table 1. FEN alone is practically insoluble in this aqueous solution. All binary systems showed better dissolution properties than the drug alone. Since there is a small concentration of dissolved cyclodextrin in the medium, the physical mixture shows increased solubility compared to the API alone, but this enhancement is limited. However, with the kneaded and co-ground products, a much greater increase in solubility was achieved. The co-ground product exhibits a faster dissolution rate and a slightly better solubility than the kneaded product. This is probably because the co-ground product is completely amorphous, while the kneaded product remained slightly crystalline.

### 3.6. In Vitro Diffusion Studies

Diffusion through dialysis bag was evaluated to further support in vitro analysis. Dialysis bags with the suspension of FEN and products were placed in phosphate buffer. Low diffused drug content can be seen in the case of the pure API, physical mixture, and kneaded product, due to the low solubility and dissolution rate. The beneficial effect of CD in the physical mixture is only seen in the second half of the diffusion test, probably due to the slow dissolution, but in the final stage, the amount of drug dissolved is comparable to the kneaded product. However, the co-ground product shows higher API content in the acceptor phase. All measured samples show a higher percentage of FEN than observed during dissolution studies due to a relatively lower active substance/release medium ratio compared to the previous measurement (Appendix A).

## 4. Conclusions

Physicochemical and in vitro methods were performed to evaluate the time course of the green preparation process of cyclodextrin complexation.

Based on the DSC thermograms with the disappearance of the melting point correlated to the API, suggesting amorphous properties, this observation was supported by XRPD measurements. According to these diffractograms, the grinding and solvent method led to amorphous products, while kneading resulted in a slightly crystalline mixture. The SEM-EDS map images showed that the granules contained fewer drug-specific signals as the grinding time progressed, which also supports the observation of possible complex formation. For the characterization of molecular relationships, the recorded FTIR spectra were evaluated by correlation analysis. The entire 60 min process showed three trends and four different spectrum sections were examined. These observed changes in spectra suggested a three-stage complex formation process, where hydrogen bonding formed between the C=O group of FEN and the -O-H groups of DIMEB during the complexation process and secondary bonds between the aromatic rings of FEN and hydrophobic internal space of the CD. According to these findings, XRPD and DSC studies may be appropriate to monitor the degree of complexation during grinding if intermolecular interactions between materials have been characterized by other methods.

Obtained products showed enhanced properties during in vitro dissolution and diffusion studies. Among all measured samples, the ground complex proved to have the highest values, and increased bioactivity in the final dosage form is anticipated in future work.

## Figures and Tables

**Figure 1 pharmaceutics-14-01329-f001:**
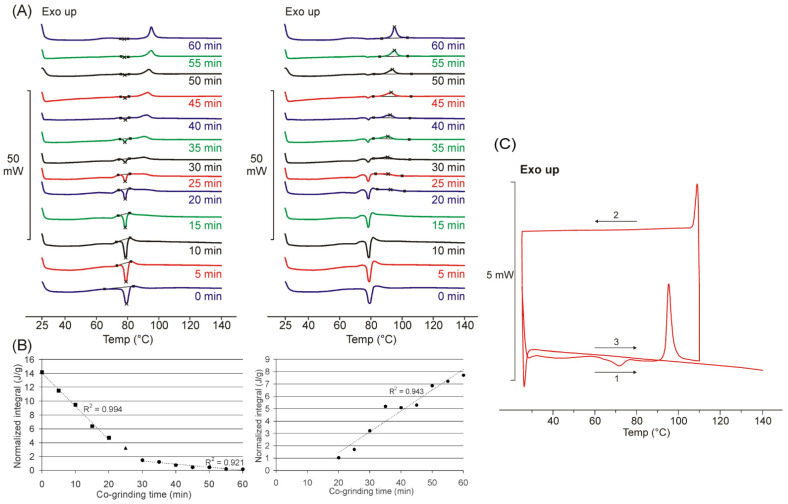
DSC curves of co-ground products show decreasing endothermic (ca. 80.5 °C) and increasing exothermic peak with the increasing grinding time, peak points were marked with × (**A**), and the corresponding normalized integral of the areas plotted as a function of grinding time (**B**). DSC result of the double heating procedure suggests a new crystalline phase with thermal stability during the second heating phase (**C**). Double heating process sub-steps were the followings: first heating (1), cooling (2), second heating (3).

**Figure 2 pharmaceutics-14-01329-f002:**
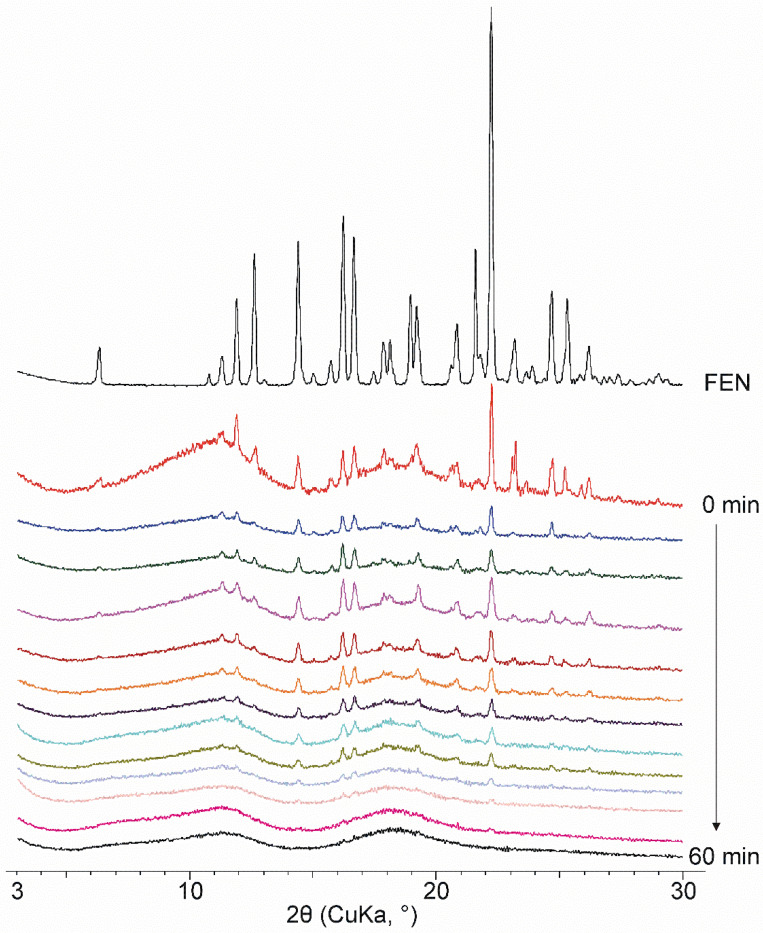
Diffractograms of FEN and co-ground products measured in every 5 min from 0 min to 60 min product. With the increasing grinding time, characteristic peaks decreased, and an amorphous product has been observed.

**Figure 3 pharmaceutics-14-01329-f003:**
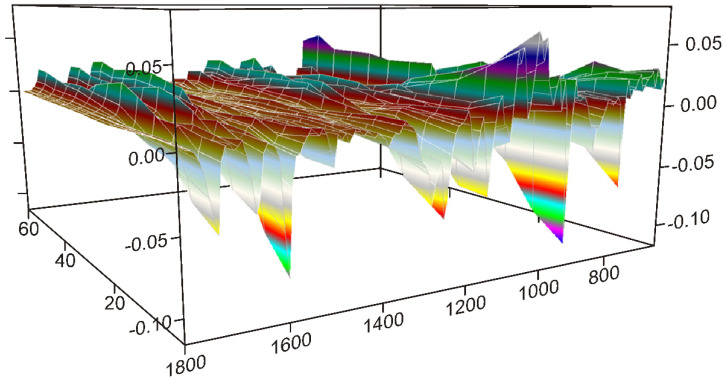
The dynamic surface (the different colors of the surface produced by the 3D representation belong to the values of the *Y*-axis), constructed from the truncated and normalized spectra between 1800 and 630 cm^−1^, and recorded between 0 and 60 min of grinding time interval, shows that the intensities of peaks change variously: increasing or decreasing or showing maximum by time. Dividing the surface into three 20-min long periods separated these intensity variations, and it gave the possibility to reveal the main processes occurring during the different stages of grinding.

**Figure 4 pharmaceutics-14-01329-f004:**
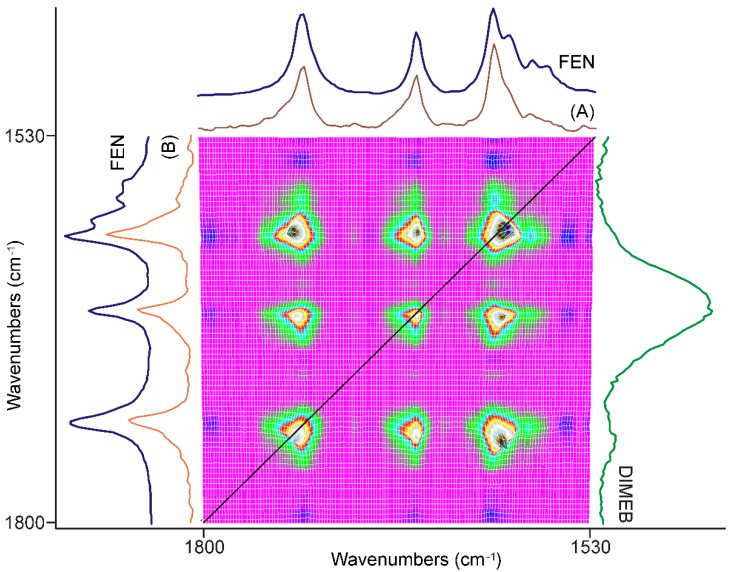
Part of the synchronous correlation surface was calculated from the spectra recorded during the first 20 min of grinding, between 1800 and 1530 cm^−1^. Diagonal peaks, marked with the black line, showed a strong positive correlation for all three peaks with each other in the region. The axial sessions taken at 1728 cm^−1^ (A) and 1651 cm^−1^ (B) showed that only the peak intensities of FEN were affected.

**Figure 5 pharmaceutics-14-01329-f005:**
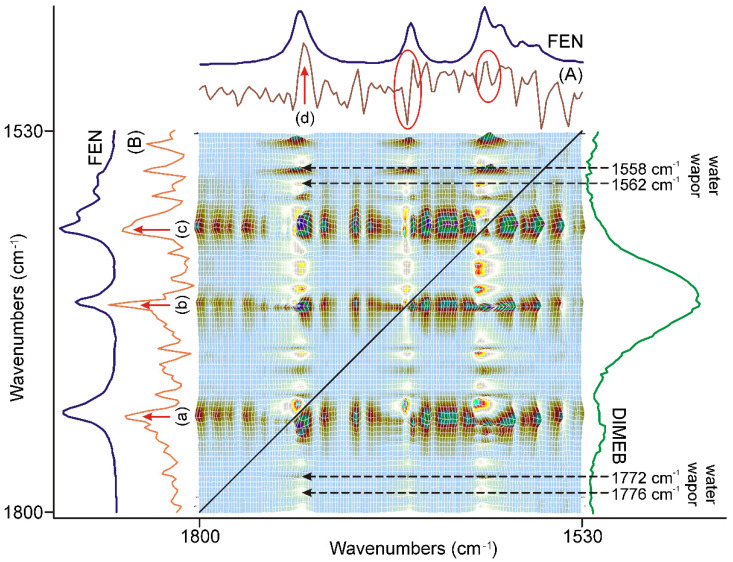
The asynchronous correlation surface of the first 20 min of grinding, between 1800 and 1530 cm^−1^, had axial symmetry to the diagonal black line. Some unexpectedly strong correlation occurred outside these regions, some marked by black broken arrows, on both sides of the diagonal line and they were attributed to the residues of water vapor bands. On the other hand, (A) the x-axial section of the surface at 1732 cm^−1^, and (B) the y-axial section of the surface at 1726 cm^−1^ indicated strong changes in the ranges of the characteristic bands of FEN: (a) at 1732 cm^−1^, (b) at 1653 cm^−1^, (c) at 1600 cm^−1^, and (d) at 1726 cm^−1^. In the case of (B), the other two characteristic bands were impossible to identify due to the interference of water vapor peaks, marked with a red ellipsis.

**Figure 6 pharmaceutics-14-01329-f006:**
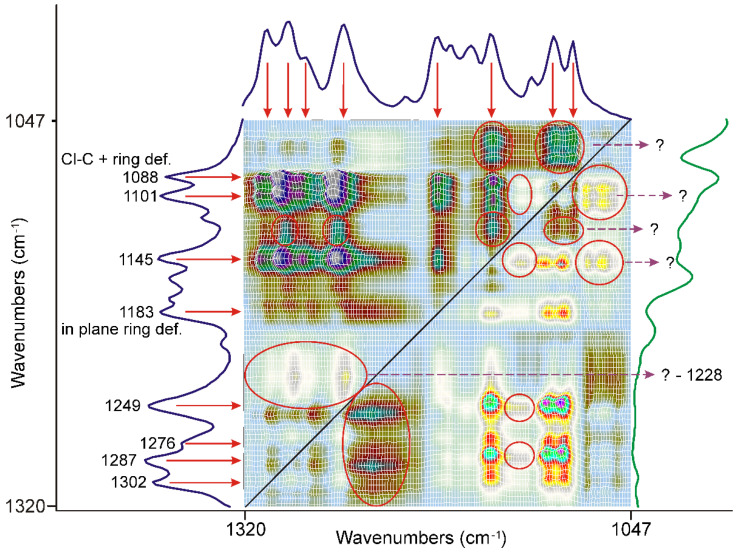
The most significant range on the asynchronous surface is outside the C=O stretching region, characteristic bands of FEN marked with red arrows. Most of the peaks, assigned to the deformation modes of the methyl groups and the C-C-O stretching modes of the ester group, and the etheric oxygen between the aliphatic part of the molecule and the non-chlorine substituted ring, showed a strong correlation to each other. Some peaks, which also had a strong correlation, were assigned to various modes of the aromatic rings, like the Cl-C stretching mode combined with the ring deformation mode at 1088 cm^−1^ or the ring in-plane bending mode of the O-substituted ring at 1183 cm^−1^ Certain broad features, which cannot be attributed either to FEN or to DIMEB, marked with a red ellipsis, purple dashed arrows and question marks.

**Figure 7 pharmaceutics-14-01329-f007:**
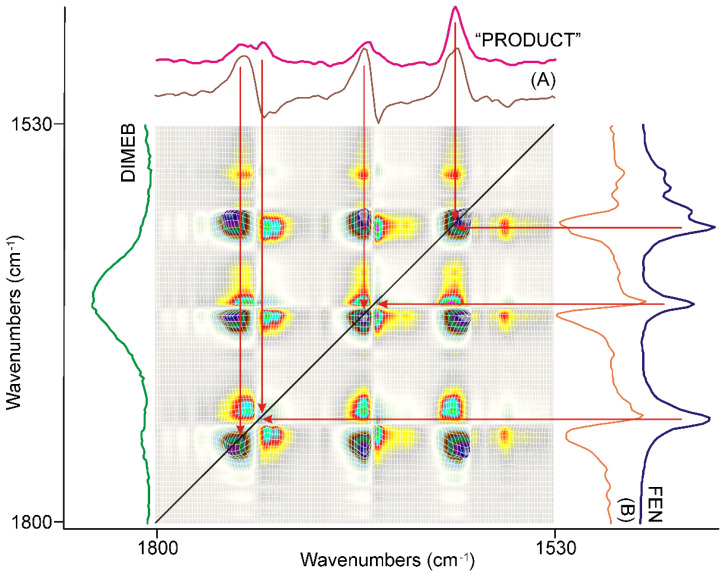
The synchronous correlation surface calculated from the spectra recorded during the second 20 min long period of time, between 1800 and 1530 cm^−1^, showed a much more complex picture than that of the previous 20 min. The axial sections were taken at 1740 cm^−1^ (A) and 1726 cm^−1^ (B), confirming that the formation of a kind of product was the dominating process, while the consumption of FEN had a lesser contribution. The spectrum of “PRODUCT” was calculated from the average of all spectra recorded during the full grinding process by consecutive subtraction of the spectra of the starting materials, FEN and DIMEB.

**Figure 8 pharmaceutics-14-01329-f008:**
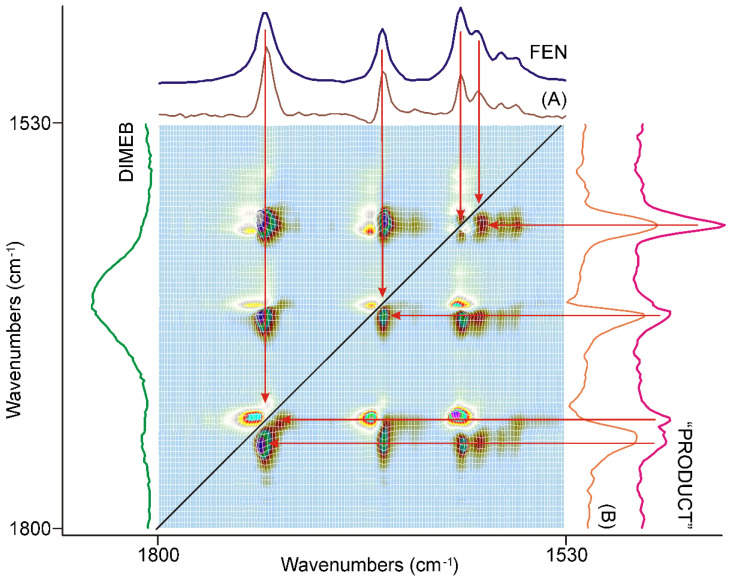
The asynchronous correlation surface calculated from the spectra recorded during the second 20 min of grinding was free of the interference of residual water vapor peaks and showed real asynchronous processes. The positions of the off-diagonal peaks were determined practically by the same wavenumbers as those of the diagonal peaks in the corresponding synchronous correlation surface. The x-axial section taken at 1740 cm^−1^ (A) and at 1728 cm^−1^ (B) showed that the positive sign of the off-diagonal peaks in the upper-left triangle were attributed to the peaks of FEN, and the same in the lower-right triangle to the peaks of “PRODUCT”, so the intensity changes in this region had an asynchronous component above the synchronous one, at the beginning of the period.

**Figure 9 pharmaceutics-14-01329-f009:**
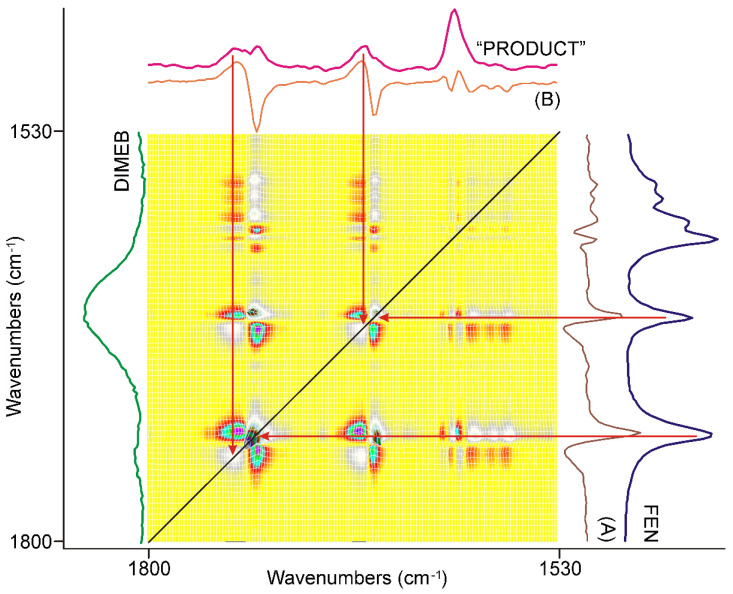
The synchronous correlation surface calculated from the spectra recorded during the last 20 min long period of time, between 1800 and 1530 cm^−1^, showed a very similar picture to that of the previous 20 min. The axial sections were taken at 1742 cm^−1^ (A) and 1728 cm^−1^ (B), confirming that the consumption of the residue of FEN was the dominating process accompanied by lesser contribution from the formation of “PRODUCT”. The range of the aromatic ring stretching modes around 1600 cm^−1^ showed very low intensities.

**Figure 10 pharmaceutics-14-01329-f010:**
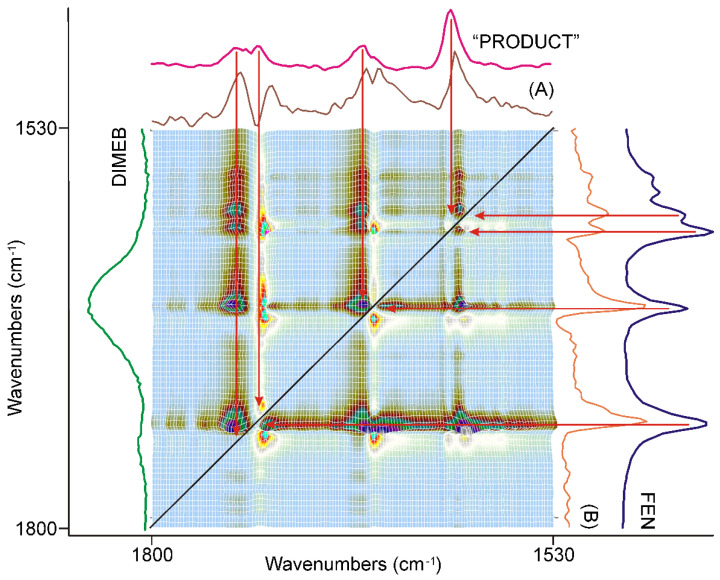
The asynchronous correlation surface calculated from the spectra recorded during the last 20 min was similar to that of the previous 20 min, except for the peaks’ switched sign. The x-axial section taken at 1728 cm^−1^ (A) and at 1740 cm^−1^ (B) showed that the positive sign of the off-diagonal peaks in the upper-left triangle was attributed to the peaks of “PRODUCT”, and the same in the lower-right triangle to the peaks of FEN, so the intensity changes in this region had an asynchronous component above the synchronous one, at the end of the period.

**Figure 11 pharmaceutics-14-01329-f011:**
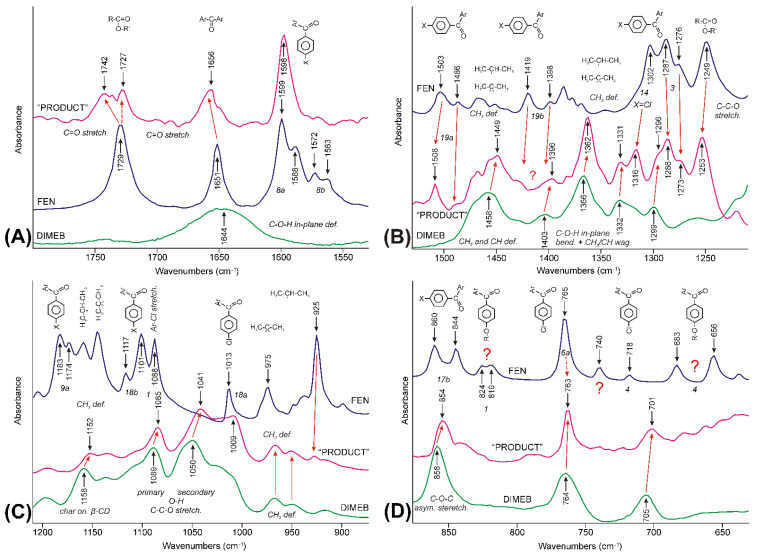
Comparison of IR-ATR spectra of FEN (purple line), DIMEB (green line), and “PRODUCT” (magenta line), in the 1800–630 cm^−1^ spectral range, which was divided into four parts (**A**–**D**). Red arrows show interpretation of the peaks of the raw materials shifted and widened. The vibrations that were impossible to identify, marked with question marks.

**Figure 12 pharmaceutics-14-01329-f012:**
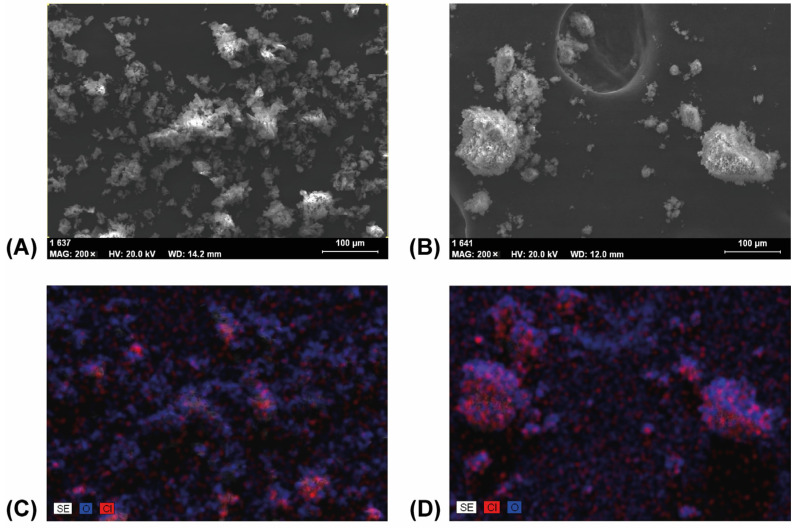
SEM images of samples taken prior to grinding (**A**) and after 20 min of grinding (**B**) showed less only DIMEB containing separated particles after grinding. The corresponding EDS mapping images (**C**,**D**) showed more evenly distributed FEN in the aggregates, with a higher ratio on the surface, after 20 min of grinding.

**Figure 13 pharmaceutics-14-01329-f013:**
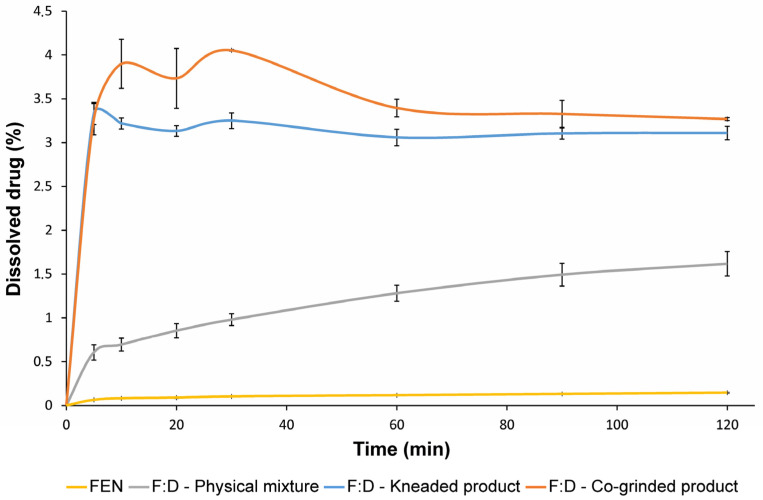
Dissolution curves of FEN, FEN:DIMEB (F:D) physical mixture, kneaded product, and co-ground product. All products show increased dissolution properties compared to pure FEN.

**Table 1 pharmaceutics-14-01329-t001:** Calculated dissolution efficiency values at 60 and 120 min of FEN, and increased values of FEN:DIMEB (F:D) products.

	DE 60 min (%)	DE 120 min (%)
FEN	0.09 ± 0.01	0.11 ± 0.01
F:D—physical mixture	0.94 ± 0.09	1.20 ± 0.12
F:D—kneaded	3.05 ± 0.10	3.07 ± 0.09
F:D—co-ground	3.59 ± 0.10	3.46 ± 0.11

## Data Availability

Not applicable.

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
