# Peer review of "Cyclodextrin Complexation of Fenofibrate by Co-Grinding Method and Monitoring the Process Using Complementary Analytical Tools"

_pharmaceutics, 2022, doi:10.3390/pharmaceutics14071329_

Round 1

Reviewer 1 Report

The presented work is a great analysis of the co-ground and kneading complexes of Fenofibrate with Cyclodextrins. The authors have focused its work on the physicochemical analysis and dissolution rate, giving a lot of results that justify the discussion and conclusion. 

However, and despite the quality of the analysis, the article lacks any biological proof of its greater effectiveness or possible loss of bioactivity due to the protocols used. In the current state, the work is more for pure physicoanalytical or chemical journals, not for pharmaceutics dealing with pharmacology.

For that reason, I reccommend a major revision where:

- I strongly suggest the authors carrying out some biotest to analyzed the bioactivity of the complexed drug.

- Revise the conclusions, because it seems a discussion. Conclusions and discussion are different points.

Author Response

Reviewer 1

The presented work is a great analysis of the co-ground and kneading complexes of Fenofibrate with Cyclodextrins. The authors have focused its work on the physicochemical analysis and dissolution rate, giving a lot of results that justify the discussion and conclusion.

However, and despite the quality of the analysis, the article lacks any biological proof of its greater effectiveness or possible loss of bioactivity due to the protocols used. In the current state, the work is more for pure physicoanalytical or chemical journals, not for pharmaceutics dealing with pharmacology.

For that reason, I reccommend a major revision where:

- I strongly suggest the authors carrying out some biotest to analyzed the bioactivity of the complexed drug.

- Revise the conclusions, because it seems a discussion. Conclusions and discussion are different points.

Answer:

Thank you very much for your comments and questions. English language and style were modified. Below are listed all responses and changes made in the paper, according to your suggestions. You can find these added or modified parts with green color in the manuscript.

I strongly suggest the authors carrying out some biotest to analyzed the bioactivity of the complexed drug.

            During the experimental work, we focused on the monitoring of physico-chemical changes, thus supporting the effectiveness of the solvent-free process used in the complexation process. Based on the reviewer’s suggestion introduction was supplemented also by diffusion studies to further support in vitro results. Therefore we could get an in vitro overview (dissolution and diffusion) about the further applicability of the composizions. However in this manuscript we did not intend to make bioactivity studies, but as the last sentence suggest in a future work a final dosage form will be formulated, and bioactivity studies will be probably performed.

Revise the conclusions, because it seems a discussion. Conclusions and discussion are different points.

            At the suggestion of the reviewer conclusion part has been changed significantly.

Reviewer 2 Report

Kondoros et al. presented a detailed analytical study of the complexation of fenofibrate by dimethyl-β-cyclodextrin. The complex was prepared using kneading and co-grinding methods. The formation of the inclusion complex was thoroughly monitored over time by means of DSC, XRPD and 2-dimensional correlation FTIR analysis, in addition to SEM observation and dissolution studies.

The work is scientifically sound and the experimental results are discussed in depth and clearly. The manuscript can be accepted after minor revision.

Line 181, should it be “… from the grains”?

Line 194, “Samples were measured spectrophotometric method at 296 nm…” a preposition is missing

Line 223, Have you tried to perform the DSC analysis of the drug alone after 60 min of grinding? Is the crystallinity affected by the grinding process?

Lines 238, 240, it is an exothermic peak, not endothermic

Line 252, Have you performed an XRD analysis to confirm whether it is a new crystalline phase?

Lines 660-664, a repetition of lines 655-659

Author Response

Reviewer 2

Kondoros et al. presented a detailed analytical study of the complexation of fenofibrate by dimethyl-β-cyclodextrin. The complex was prepared using kneading and co-grinding methods. The formation of the inclusion complex was thoroughly monitored over time by means of DSC, XRPD and 2-dimensional correlation FTIR analysis, in addition to SEM observation and dissolution studies.

The work is scientifically sound and the experimental results are discussed in depth and clearly. The manuscript can be accepted after minor revision.

Line 181, should it be “… from the grains”?

Line 194, “Samples were measured spectrophotometric method at 296 nm…” a preposition is missing

Line 223, Have you tried to perform the DSC analysis of the drug alone after 60 min of grinding? Is the crystallinity affected by the grinding process?

Lines 238, 240, it is an exothermic peak, not endothermic

Line 252, Have you performed an XRD analysis to confirm whether it is a new crystalline phase?

Lines 660-664, a repetition of lines 655-659

Answer:

Thank you very much for your comments and questions. Below are listed all responses and changes made in the paper, according to your valuable suggestions. You can find these added or modified parts with blue color in the manuscript.

Line 181, should it be “… from the grains”?

Line 194, “Samples were measured spectrophotometric method at 296 nm…” a preposition is missing

            Based on the reviewer’s suggestions correction was applied in these cases.

Line 223, Have you tried to perform the DSC analysis of the drug alone after 60 min of grinding? Is the crystallinity affected by the grinding process?

            We did not performed this experiement, but we are planning to write another article in this topic, where we will compare the API to products in this aspect.

Lines 238, 240, it is an exothermic peak, not endothermic

            Thank you for your comment, it has been corrected.

Line 252, Have you performed an XRD analysis to confirm whether it is a new crystalline phase?

            Thank you for your insightful remarks. Yes, we have performed this experiement, but since the article has become essentially long, a subsequent manuscript will contain these measurements, which we would like to publish in the near future, along with the previously mentioned grinding of the active ingredient alone.

Lines 660-664, a repetition of lines 655-659

            Thank you for your comment, it has been corrected. The conclusion section was revised.

Reviewer 3 Report

Dear authors

You have done extensive amount of work in your paper and it is well written.

However I have a few suggestions:

In introduction DIMEB abrevation shod be introduced and you should explain why this CD is chosen

The aim should be without citation and more clearly stated

Result and discussion is well done but there is lack of concerete explenation what does in physical sence each change in section of 20 minutes means from formulation aspect and some references could be added whate possible could be aded

Why dissolution is not done after co-grinding time period that were analysed i.e. diss after 20 min 40 min 60 mi

Conclusion should be without refernces and more concise

Pleas think about title change since I do believe presebt titel is too general.

Best of luck and keep with a good work

P.s. title should contain something about cogrinding time correlation to physical state of mixture

Author Response

Reviewer 3

Dear authors

You have done extensive amount of work in your paper and it is well written.

However I have a few suggestions:

In introduction DIMEB abrevation shod be introduced and you should explain why this CD is chosen

The aim should be without citation and more clearly stated

Result and discussion is well done but there is lack of concerete explenation what does in physical sence each change in section of 20 minutes means from formulation aspect and some references could be added whate possible could be aded

Why dissolution is not done after co-grinding time period that were analysed i.e. diss after 20 min 40 min 60 mi

Conclusion should be without refernces and more concise

Pleas think about title change since I do believe presebt titel is too general.

Best of luck and keep with a good work

P.s. title should contain something about cogrinding time correlation to physical state of mixture

Answer:

Thank you very much for your comments and questions. English language and style were modified. Below are listed all responses and changes made in the paper, according to your suggestions. You can find these added or modified parts with yellow color in the manuscript.

In introduction DIMEB abrevation shod be introduced and you should explain why this CD is chosen

            Based on the reviewer’s suggestion introduction was supplemented by DIMEB abbreviation and explanation of choosing this CD-derivative.

The aim should be without citation and more clearly stated

            The text of the cited article has been moved within the introduction and reworded to make it easier to interpret.

Result and discussion is well done but there is lack of concerete explenation what does in physical sence each change in section of 20 minutes means from formulation aspect and some references could be added whate possible could be aded

            Thank you for your remarks. Conclusion have been modified for further explanation of changes in 20 minutes sections.

Why dissolution is not done after co-grinding time period that were analysed i.e. diss after 20 min 40 min 60 min

            As final and most effective product we only used 60 minute product in dissolution studies, since other products contained residual crystalline drug. For this reason this would be used in the preparation of a pharmaceutical product for possible future use.

Conclusion should be without refernces and more concise

            At the suggestion of the reviewer conclusion part has been changed significantly.

Pleas think about title change since I do believe presebt titel is too general.

            Thank you for your suggestion, based on this title has been modified:

Cyclodextrin complexation of fenofibrate by co-grinding method and monitoring the process using complementary analytical tools

Round 2

Reviewer 1 Report

Although the authors have considerably improved the article, I still think that the lack of any evidence on the activity of the samples is necessary to be considered in pharmaceutics, I recommend resubmission to another Journal.

Author Response

Dear Reviewer,

thanks for your opinion about our work. Sorry but we have no possibility at this moment to perform the suggested in vivo test.